# Interaction of Tryptophan- and Arginine-Rich Antimicrobial Peptide with *E. coli* Outer Membrane—A Molecular Simulation Approach

**DOI:** 10.3390/ijms24032005

**Published:** 2023-01-19

**Authors:** George Necula, Mihaela Bacalum, Mihai Radu

**Affiliations:** 1Department of Computational Physics and Information Technologies, Horia Hulubei National Institute for R&D in Physics and Nuclear Engineering, 077125 Magurele, Romania; 2Department of Life and Environmental Physics, Horia Hulubei National Institute for R&D in Physics and Nuclear Engineering, 077125 Magurele, Romania

**Keywords:** antimicrobial-peptide, lipopolysaccharide, MD, PMF, outer membrane, *E. coli*

## Abstract

A short antimicrobial peptide (AMP), rich in tryptophan and arginine (P6—HRWWRWWRR-NH2), was used in molecular dynamics (MD) simulations to investigate the interaction between AMPs and lipopolysaccharides (LPS) from two *E. coli* outer membrane (OM) membrane models. The OM of Gram-negative bacteria is an asymmetric bilayer, with the outer layer consisting exclusively of lipopolysaccharide molecules and the lower leaflet made up of phospholipids. The mechanisms by which short AMPs permeate the OM of Gram-negative bacteria are not well understood at the moment. For this study, two types of *E. coli* OM membrane models were built with (*i*) smooth LPS composed of lipid A, K12 core and O21 O-antigen, and (*ii*) rough type LPS composed of lipid A and R1 core. An OmpF monomer from *E. coli* was embedded in both membrane models. MD trajectories revealed that AMP insertion in the LPS layer was facilitated by the OmpF-created gap and allowed AMPs to form hydrogen bonds with the phosphate groups of inner core oligosaccharides. OM proteins such as OmpF may be essential for the permeation of short AMPs such as P6 by exposing the LPS binding site or even by direct translocation of AMPs across the OM.

## 1. Introduction

Lipopolysaccharide (LPS) molecules are an essential component of the outer membrane (OM) of Gram-negative bacteria. The OM of Gram-negative bacteria is an asymmetric bilayer, with the outer leaflet consisting exclusively of lipopolysaccharide molecules and the lower leaflet made up of phospholipids. A typical LPS molecule consists of three distinct subunits: (*i*) lipid A (endotoxin), a glucosamine-based phospholipid with a conserved structure; (*ii*) the “core”, a non-repeating oligosaccharide, and (*iii*) O-antigen, a distal polysaccharide consisting of 1 to 40 repeating units [1,2]. The LPS molecules form a protective barrier against antibiotics and other therapeutic substances such as antimicrobial peptides (AMP) [1,3] via two mechanisms: (*i*) electrostatic linking negatively charged 3-Deoxy-D-manno-octulosonic acid (Kdo) moieties of “core” oligosaccharide chains, via divalent cations (Ca^2+^ and Mg^2+^) [4,5], and (*ii*) much tighter packing of acyl chains than that found in bilayers composed of typical phospholipids [5]. Transport across the OM is regulated by the outer membrane proteins (OMP) it contains, and a tight interaction between OMPs and the LPS layer is essential for OM impermeability [6]. *E. coli* outer membrane protein F (OmpF) is the best studied and the most abundant porin (~10^5^ copies per cell) [7]. OmpF is a homotrimer consisting of three 16-stranded β-barrels connected by eight loops (L1–L8) on the extracellular side, and eight reverse turns on the periplasmic side [8]. A long loop (L3) runs along one side of each lumen and forms a constriction region [9]. The contact between the three monomers is stabilized by hydrophobic and polar interactions by loop 2 (L2), which folds back outward and latches the neighboring monomer [10]. OmpF porin simultaneously transports water molecules, positive ions, and negative hydrophilic molecules with a mass of up to 600 Da across the outer membrane [8,11]. In spite of the stringent molecular mass restriction, it was shown that HPA3P peptide (2.9 kDa), an analogue of HP (2–20) antimicrobial peptide, is able to interact with the constriction loop (L3) of OmpF, which lead to individual monomer blocking and visible peptide translocation events across the OM [12]. Surprisingly, even protamine, a 32 amino acid polycationic peptide (5.1 kDa) is able to permeate the bacterial OMPs CymA from *Klebsiella oxytoca* and OmpF from *E. coli* [13,14].

Typical AMPs are small cationic and amphipathic peptides that have well-defined hydrophilic and hydrophobic regions [15,16]. The hydrophobicity level of cationic AMPs correlates well with their ability to traverse the LPS layer, and implicitly with their antimicrobial activity [17]. The mechanisms by which antimicrobial peptides permeate the outer membrane of Gram-negative bacteria are poorly understood due to the complexity and molecular variability of LPS molecules [18]. Several studies have suggested that the attachment of AMPs to the OM is followed by increased mobility and disordered packing of O-antigen and acyl chains of LPS, which determines the expansion or “loosening”, as well as depolarization of the outer leaflet [19,20]. This mechanism of action produces a transient “self-promoted-uptake” similar to the “carpet-like” mechanism proposed for the destabilization of phospholipid bilayers by AMPs [21]. It was suggested that a wide range of AMPs such as CAP18, polymyxin B, indolicidin, etc., are able to neutralize LPS molecules by direct binding and to even prevent septic shock [22].

In this study, a short (nine amino acid) peptide, rich in arginine and tryptophan residues (P6—HRWWRWWRR-NH2), was used in various molecular dynamics (MD) simulations to probe the interactions between AMPs and both types of LPS molecules—smooth LPS (S-LPS) and rough LPS (R-LPS)—built into distinct *E. coli* membrane models (Figure 1b,d). An OmpF monomer from *E. coli* was embedded in both membrane models (Figure 1a,c). The P6 peptide proved to have good efficacy against both Gram-negative and Gram-positive bacteria [23]. Two independent variables were introduced in the form of two molecular derivatives of the P6 peptide, which were subjected to MD simulations in the same conditions as P6: (*i*) addition of an arginine residue to the N-terminus (HRWWRWWRRR-NH2), designated P6m and (*ii*) inversion of hydrophobic and hydrophilic regions to create a deconstructed version of the P6 peptide (HWRRWRRW-NH_2_), designated Pxm (Table 1). The former was used to assess whether the binding mode of small AMPs to the LPS layer is impacted by a single amino acid difference, while the latter was used to observe the impact of inverting the hydrophobic and hydrophilic regions on the binding mode. MD trajectory analysis revealed a putative binding site of the P6 and derivative peptides represented by phosphate groups of inner core oligosaccharide of S-LPS and R-LPS. AMPs binding to the LPS binding site was facilitated by the gap in the LPS layer formed by OmpF. Finally, due to the proximity of the putative LPS binding site to the OmpF pore opening, the possibility of AMPs translocation through OmpF porin was assessed with the help of steered MD simulations.

## 2. Results

### 2.1. AMPs Binding to the S-LPS E. coli Membrane Model

During MD simulations of the three AMPs in interaction with the S-LPS *E. coli* membrane model, only about a third of peptide molecules entered in contact with the LPS layer. Only three P6 molecules interacted in any meaningful way with the O-antigen polysaccharide chains, while a single molecule managed to reach the inner core oligosaccharide and even interact with Hep I (l-glycero-d-manno-heptose) residues, at a depth of ~10 Å under the LPS surface—calculated as the average center of mass (COM) of β-Gal II residues of O21 O-antigen. This P6 molecule was chosen as the starting configuration for the non-equilibrium simulations in order to calculate the potential of mean force (PMF). In this configuration, the P6 peptide forms most of the hydrogen bonds with the Hep I, Hep II residues, especially with the phosphate group, mainly with R5 and R8 residues. The R8 residue of the P6 peptide is also interacting, although to a lesser extent, with the Glc III (d-glucose) and Gal I (d-galactose) residues from the upper core oligosaccharide. The dissociation of the P6 peptide from the LPS leaflet, during the non-equilibrium simulations, was achieved in two stages: (*i*) the progressive rupture of the hydrogen bonds formed between the R5 and R8 residues of the P6 and the Hep I, Hep II, Gal I, and Glc III residues from the core, as a consequence of dislocating the P6 COM by 7 Å, and (*ii*) forming and subsequent breaking of hydrogen bonds with β-GalNAc (N-acetyl-d-glucosamine) or β-GlcNAc (N-acetyl-d-galactosamine), and finally with the terminal galactose residues (β-Gal I & II) from the O21 O-antigen subunit, as the peptide is completely extracted from the LPS layer. Complete detachment of P6 from the LPS molecules was achieved after displacing the peptide COM by 20 Å from the initial position (Figure 2a). The binding energy minima of the P6 peptide interacting with S-LPS molecules was calculated at −13.99 kcal/mol at 10.66 Å (z axis) under the LPS surface. The two-stage binding/dissociation of the P6 peptide from the LPS layer is also indicated by the analysis of average hydrogen bonds in the non-equilibrium simulations (Figure 2b).

The MD simulation of the P6m peptide with the S-LPS *E.coli* membrane model was the most eventful, with multiple P6m molecules inserted early on in the LPS layer at a depth of >10 Å, and a single molecule even interacted with OmpF, at a depth of ~15 Å under the LPS surface. The binding energy profile of the P6m peptide interacting with the core oligosaccharides, O-antigen chains, and even with some residues of OmpF indicated the strong interaction of this AMP with the S-LPS *E. coli* membrane model. The protein–protein interaction between P6m and OmpF is limited to the loops L5 and L6: H1 residue of P6m forms a hydrogen bond with the side chain of Q203 located on loop 5 (L5) of OmpF; R2 residue forms two hydrogen bonds with side chain atoms of N246 from loop 6 (L6); while W4 forms a hydrogen bond with the main chain O atom of F244, also from L6 of OmpF (Figure 3d). Structural stability of the OmpF in all MD simulations was analyzed by root mean-square deviation (RMSD) of all atoms, those of the β-barrel, and of the key loops (Appendix A), and by root mean-square fluctuation (RMSF) of all amino acids (Appendix A). As in the case of the non-equilibrium simulations performed with the P6 peptide, the dissociation of the P6m peptide from the LPS binding site was also achieved in two stages: (*i*) breaking of the hydrogen bonds between the triple N-terminus R residues and Hep I and II phosphate groups, as a result of displacing P6m COM by 5 Å, and (*ii*) the successive forming and breaking of hydrogen bonds with the upper K12 core residues (e.g., Gal I) and with the O-antigen (β-GlcNAc and β-Gal II). The complete detachment of the P6m peptide from the LPS leaflet was achieved after displacing the peptide COM by 30 Å. The binding energy minima of P6m interacting with the LPS leaflet is −33.6 kcal/mol at 12.32 Å (z axis) under the LPS surface. The strong binding energy is also indicated by the high number of hydrogen bonds formed with the LPS molecules.

The impact of increasing the number of O-antigen subunits on the ability of P6m peptide to permeate the LPS leaflet was evaluated with the help of a molecular system built with LPS molecules composed of five repeating O21 O-antigen units. The analysis of MD trajectories revealed only limited (1–2 Å) and short duration insertions of P6m peptide molecules in the LPS layer. The P6m peptide established contacts exclusively with residues from the terminal O-antigen unit. In the starting configuration used for non-equilibrium simulations, the P6m peptide is interposed between four long polysaccharide chains and forms stable hydrogen bonds with all R residues and some W residues, mainly with the terminal Gal and Glc residues from the terminal O-antigen unit. The complete detachment of P6m around 20 Å is explained by the progressive detachment from the N-terminus to the C-terminus region, and also by the presence of the additional R residue. The binding energy minima of P6m peptide in interaction with five repeating O21 O-antigen subunits is −9.43 kcal/mol at 7.67 Å (z axis) above the LPS surface (Figure 2a).

In a similar manner, only two of the 12 modified Pxm peptides were superficially inserted (1–2 Å) in the LPS layer during MD simulation, despite only interacting with a membrane model with a single O21 O-antigen unit. The Pxm peptide establishes a limited number of hydrogen bonds with low occupancy, mainly with the terminal galactose residues (β-Gal I or β-Gal II), but also with the β-GlcNAc residue of O21 O-antigen, which was reflected in the binding energy minima of −1.32 kcal/mol at 8.12 Å (z axis) above the LPS surface (Figure 2a).

An S-LPS *E. coli* membrane model built with five repeating O21 O-antigen units, in conjunction with 12 P6 peptides, which was converted to a coarse-grained (CG) MARTINI system, was employed to test the hypothesis of AMPs direct permeation of LPS layer. Despite the extended CG simulation time (145 μs), only three P6 molecules were inserted between the simplified representations of the O-antigen chains. The first P6 peptide is inserted after 10 μs, in a relatively stable configuration at a depth of 10 ± 3 Å, which is maintained throughout the simulation. The second peptide insertion is recorded after 45 μs, and reaches a maximum insertion level of approximately 4 Å, after another 35 μs. The last insertion occurred late in the simulation (115 μs), and reached a maximum depth of 5–7 Å. Although the CG simulation managed to replicate the atomistic insertion results of the P6 peptide, no P6 molecule came close to the insertion levels observed for the P6m peptide in the atomistic S-LPS *E. coli* membrane model (Appendix A).

### 2.2. AMPs Binding to the R-LPS E. coli Membrane Model

MD simulations of the three AMPs in contact with the R-LPS *E. coli* membrane model shared the sparse interactions with the LPS layer observed for the S-LPS *E. coli* membrane model. Only three to four peptide molecules interacted with the LPS surface, but in each MD simulation there was at least some level of interaction between OmpF and AMPs. This change was facilitated by the gap in the LPS layer created by the transmembrane channel and especially by the significantly shorter LPS molecules without an O-antigen component.

A single P6 peptide entered the gap left in the LPS layer during MD simulation and interacted with both OmpF channel residues and LPS inner core oligosaccharides. In the configuration used to calculate the PMF profile, the P6 peptide forms hydrogen bonds mainly with the phosphate groups of Hep I and Hep II residues from the R1 inner core and Glc I residue located in the outer core. The P6 peptide is interacting with a fairly low occupancy hydrogen bond formed between the H1 residue and the side chain O atom of E29 (L1) of OmpF. As the P6 peptide is displaced from the binding site in the non-equilibrium simulations, the initial hydrogen bonds are gradually broken and new ones form mainly with Gal I and β-Glc from the outer core. The binding energy minima of P6 in interaction with the S-LPS *E. coli* membrane model is −10.4 kcal/mol at 0.57 Å (z axis) under the LPS surface. The two P6 derivative peptides have interacted in a similar manner with R1 core residues but differed in level of insertion in the LPS layer and OmpF interacting residues. The P6m peptide forms hydrogen bonds with E284, G285, and I286 residues located on loop 7 (L7) of OmpF, while the Pxm peptide is in contact with N27 and E29, located on loop 1 (L1), and K243 and N246 residues located on loop 6 (L6). Even the PMF profiles of the two modified peptides were quite similar, with P6m binding energy minima of −6.2 kcal/mol at 0.11 Å (z axis) under the LPS surface, while the Pxm peptide binding energy minima of −7.32 kcal/mol at 2.74 Å (z axis) under the LPS surface. Essential PMF results of AMPs binding to both types of *E. coli* OM are provided in Table 2.

### 2.3. OmpF—LPS Interface Analysis

Because of the binding mode exhibited by all AMPs analyzed, in the R-LPS *E. coli* membrane model systems, at the upper level of OmpF—LPS interface, we also analyzed the possibility that AMPs could employ this mechanism to permeate the OM. Thus, the impact of OmpF on the lateral packing of lipid A and phospholipids: PPPE and PVPG was measured for the final 100 ns of all simulations by the use of area per lipid (APL) analysis. An overall reduction of area per lipid of lipid A was observed in all the simulations, with a reduction from 186.26 ± 0.86 Å^2^ to an average 161.73 ± 1.86 Å^2^ for the S-LPS *E. coli* membrane model simulations and to an average 173.96 ± 1.85 Å^2^ for the R-LPS *E. coli* membrane model simulations. Lipid A molecules within a radius of 5 Å around the OmpF exhibited a significant reduction of APL to just 115.48 ± 1.86 Å^2^ (Figure 4a) for the former LPS type and to 140.11 ± 4.62 Å^2^ for the latter (Figure 4b). The two main phospholipids (PPPE and PVPG) exhibited a similar reduction of APL, from 67.46 ± 0.73 Å^2^ to 61.36 ± 0.60 Å^2^ for PPPE and from 72.97 ± 2.82 Å^2^ to 64.66 ± 1.21 Å^2^ for PVPG in S-LPS *E. coli* membrane model simulations. A similar APL reduction was measured for the rough type LPS *E. coli* membrane model simulations to 64.36 ± 0.7 Å^2^ for PPPE and 64.23 ± 1.62 Å^2^ for PVPG. The reduction of APL was significantly larger for the lipids in direct contact with the OmpF channel, to 46.31 ± 2.09 Å^2^ for PPPE and 50.71 ± 3.01 Å^2^ for PVPG in the S-LPS *E. coli* membrane model simulations (Figure 4c), and 53.80 ± 1.83 Å^2^ for PPPE and 51.85 ± 2.83 Å^2^ in R-LPS *E. coli* membrane model simulations (Figure 4d). Due to the low number of PVCL2, the APL analysis was not performed for this type of phospholipid.

### 2.4. Dynamics of S-LPS in Interaction with AMPs

A broader conformational distribution of O-antigenic head groups was observed during the two MD simulations of the systems without a transmembrane protein, even greater than the distribution of O-antigen subunits that formed the edge of the gap in the molecular systems with an embedded OmpF channel. O-antigen polysaccharide chains tended to bend or tilt to a greater extent, which led to the formation of a region with a higher density of O-antigen units centered around a single P6 molecule for ~250 ns, and implicitly formed an area with a lower density of O21 chains, which facilitated the insertion of a P6 peptide to a greater extent than in the LSP layer surrounding the OmpF porin in the other systems (Figure 5a,b). The P6 peptide interacting with the region of LPS layer with the higher density of O-antigen units establishes contacts mainly with the two terminal β-Gal residues and to a lesser degree with β-GlcNAc, while the P6 molecule inserted in the region of LPS layer with the lower density of O-antigen units is interacting with a much more diverse set of saccharide residues: Gal I, Glc III and Hep IV form the upper core oligosaccharide and with all O-antigen residues. The binding energy minima of the former P6 molecule is −5.51 kcal/mol at a distance of 8.85 Å (z axis) above the LPS layer, while for the latter is −7.7 kcal/mol at 6.08 Å (z axis) under the LPS surface. The dynamics of the polysaccharide chains from the LPS areas with two distinct densities of O-antigen subunits was monitored by APL-like analysis, i.e., area calculated for saccharide residues. Due to the tilting or bending of LPS molecules, the Glc III residue from the upper core oligosaccharide was chosen as the reference position for consistent APL measurements. There was a slight increase of the overall APL from 179.39 ± 1.70 Å^2^ to 186. 30 ± 0.83 Å^2^ in comparison to the reference simulation, i.e., without OmpF channel and AMPs. A reduction in APL was measured for the LPS molecules in contact with P6_1 peptide (higher O21 density) to 151.01 ± 6.44 Å^2^, and was further reduced to 118.91 ± 11.58 Å^2^ at β-Glc residues, and to 105.84 ± 19.07 Å^2^ by measuring APL at β-Gal II residues. The area with a low density of O-antigen units, which was the entry point into the LPS layer for P6_2 molecule, is formed by saccharide chains with an APL of 240. 61 ± 5.9 Å^2^ at the reference position (Glc III), but it is reduced due to the severe bending or tilting of the chains to 216.61 ± 18.50 Å^2^ calculated at β-Glc level, and to 143.04 ± 22.38 Å^2^ calculated at β-Gal II level (Figure 5c).

### 2.5. Translocation of AMPs across OmpF Pore

Due the increased lateral packing of LPS molecules surrounding the OmpF channel, the path of least resistance for OM permeation of small AMPs, already inserted in the LPS layer gap formed by OmpF, seems to be more probable across the transmembrane protein, rather than through the OMP-LPS interface or by direct LPS permeation. This possibility was assessed by the force required for all three AMPs to permeate the OmpF pore, in a linear configuration, using steered MD simulations. The force profiles for all three peptides permeating the OmpF pore during steered MD simulations were similar, with a steady increase of the pulling force required for the translocation, with a peak between 2300–2700 pN, and immediately followed by a sharp decline associated with the breaking of AMPs–OmpF interaction (Figure 6). The maximum force required to translocate the AMPs across the OmpF pore and the force profile shape seems to be contingent on the constriction loop (L3) position within the OmpF lumen, which can obstruct the pore to various degrees. This finding was indicated by one of the force profiles obtained for the P6m peptide (Figure 6c, black curve), which exhibited minimum interaction with the constriction loop (L3), and required an average force of only 784 ± 138 pN (plateau phase between 15–40 Å) to translocate across the OmpF pore, while the typical peptide translocation required >2000 pN to displace the L3 loop to a less obstructing configuration. In order to reproduce this force profile, additional steered MD simulations were performed from slightly different P6m starting positions, but only an intermediate between the two force profile types was obtained, mainly because of the increased interactions with the constriction loop (L3). The force required for translocation of P6m peptide through the OmpF lumen in the additional steered MD simulation reaches two plateau phases: the first at an average 1525 ± 78 pN, followed by a quick transition to the second at an average of 833 ± 86 pN (Figure 6c, blue curve). The two combined translocation phases closely overlap the extension of the single plateau phase of the force profile resulting from the very sparse interaction between the P6m and loop L3. While the RMSD analysis results of the L3 loop during steered MD simulations are not always in agreement with the force profiles, they revealed that the P6m peptide induces the displacement of L3 residues by only ~0.5 Å (Figure 6f, black curve) while requiring only ~800 pN for OmpF translocation. Inconsistencies between the L3 loop displacement and the force required to translocate the AMPs across the pore suggest the interaction with additional elements of OmpF porin (Figure 6e,g, black curve).

## 3. Discussion

The binding of the three AMPs to both types of LPS molecules, aided by formation of a gap in LPS layer by the embedded OmpF porin, can be viewed in reverse order to describe the actual binding path: (*i*) binding of AMPs to a temporary binding site, represented by the terminal β-Gal residues of O21 O-antigen and Gal II/β-Glc of R1 outer core oligosaccharide, respectively, followed by (*ii*) the binding of the two to three successive arginine residues from the N-terminus of the P6 or P6m peptides to the actual LPS binding site, i.e., the phosphate groups of Hep I and II from the inner core oligosaccharides. These findings are supported by the suggestion that hydroxylation, diacylation, or addition of phosphoethanolamine to the LPS phosphate groups induces resistance to polymyxin E (colistin) peptides [24]. Moreover, the monoclonal antibody WN1 222-5, which is cross-reactive against many serotypes of *E. coli*, *S. enterica*, and some *Citrobacter*, *Enterobacter*, *Klebsiella* isolates [25], was used to identify the inner core side-chain heptose and the 4-phosphate on the branched heptose as the main determinants of the LPS epitope [26]. Although all three AMPs form hydrogen bonds mainly through arginine residues with LPS molecules, these contacts are not exclusive: histidine residue also forms hydrogen bonds with phosphate groups of Hep I and II, but with fairly low occupancy (1–3%).

Further evidence suggests that the inner core phosphate groups are just a part of the actual LPS binding site. The murine monoclonal antibodies S1 resulted from immunizations with *E. coli* deep rough (Re mutant) strain F515 [27], and A6, resulted from immunizations with mutant strain of *E. coli* 0111:B4 (Rc mutant) [28] recognized the lipid A by a wide range of molecular interactions, e.g., direct hydrogen bonds, bridged water interactions, salt bridges, hydrophobic contacts and week electrostatic bonds, mainly with the bisphosphate backbone [29]. The bisphosphate group of the lipid A is recognized in the TLR4–MD-2 complex by forming hydrogen bonds with positively charged residues in toll-like receptor 4 (TLR4) and myeloid differentiation factor 2 (MD-2) [30]. TNF-α production can be increased in peripheral mononuclear cells by high mobility group box 1 (HMGB1) binding and transfer of LPS molecules, which is facilitated by the interaction of the four phosphate groups of lipid A and inner core with basic patches of HMGB1 in a murine model [31]. Apart from the sporadic interaction between AMPs with one of the Kdo residues of K12 or R1 core oligosaccharides, no contact between AMPs and lipid A molecules was observed in any of the MD simulations. Almost all of the remaining eight to nine AMP molecules that did not interact with the LPS layer in each simulation formed aggregates of two or three molecules that could bind multiple released LPS molecules (endotoxins), potentially even stronger due to the exposed phosphate groups of lipid A. While some AMPs such as the previously-mentioned indolicidin, a 13-residue rich in tryptophan and proline AMP, which was shown to be effective in the reduction of endotoxin and prevention of TNF-α upregulation in a rat model [32], it seems that tryptophan- and arginine-rich antimicrobial peptides have no endotoxin neutralization activity [33,34]. The capacity of cationic AMPs to bind endotoxins, regardless of neutralization activity, may serve as a stepping stone in the search for clinical treatment of sepsis through peptide design improvement [35].

It is unclear to what extent the additional arginine amino acid residue from the P6m peptide contributes to the permeation of the LPS layer, but as a charged amino acid it does lead to a significant increase of molecular contacts with all LPS molecular components and especially with the inner core oligosaccharide phosphate groups, which may explain the considerably lower binding energy minima than either P6 or Pxm regardless of LPS type. The effects of inverting the arginine and tryptophan amino acid residues of Pxm peptide are hard to assess from the analysis of the two MD trajectories, as no Pxm peptide has inserted in the hydrophobic region of the bilayer, where the sequence alteration should have the greatest impact. In light of MD trajectory analysis in which AMPs were unable to pass beyond the terminal O-antigen saccharide residues and APL analysis overall findings for both OM models, the insertion of the three peptides into the membrane (Lipid A—PL medium) was not explored further.

The APL analysis results may invalidate the possibility that small AMPs could permeate the OM of Gram-negative bacteria by entering the gap left in the LPS layer by OMPs and use the OMP-LPS interface as entry point. Densely packing of LPS molecules by overexpressed porins in mutant Gram-negative strains which leads to a rigid OMP-LPS interface was observed by electron microscopy [36] and atomic force microscopy [37]. Moreover, it was suggested that the LPS binding site on the OmpF, consisting of basic amino acid residues that interact with the phosphate moieties of Lipid A core, is essential for stable trimer formation of the OmpF porin [6].

The presence of lipopolysaccharide, flagella, and pili on the outer membrane of *E. coli*, can mask OM porin proteins and can lead anti-rOmpF sera to not reach and recognize recombinant rOmpF protein [38]. This finding is supported by the results of this paper, in which the LPS binding site (phosphate groups) and OmpF poring were not easily accessible to AMPs, especially for S-LPS *E. coli* serotype with five O-antigen repeating units. The interactions of the P6m peptide with LPS molecules built with five O21 O-antigen units were limited mainly to contacts with Gal and Glc residues from the terminal O-antigen, despite the gap formed in the LPS layer by OmpF, which is explained by the extensive bending and tilting of the very long and flexible O-antigen chains. In contrast, the same gap in the LPS layer built with a single O21 O antigen unit enabled the insertion and the strong interaction of the P6m peptide with the phosphate groups from the inner core and even with the OmpF porin. This finding is supporting the suggestion that the LPS gap formed by the OmpF porin is required for the binding of antibacterial toxin colicin N (colN) to the exposed inner core of LPS molecules [39]. Colicin N and colicin derivatives also directly bind to OmpF, which is used both as a receptor and translocator [40]. Wild-type OmpF-colN complexes reduced the formation of OmpF-LPS complexes in vitro, suggesting that tightly bound colN to the OmpF β-barrel displaces the LPS molecules around the porin, which in turn increases the possibility of colicins translocation across OM at the protein-LPS interface [41].

The binding of the AMPs to the LPS inner core oligosaccharides phosphate groups, in the proximity of the OmpF pore opening, could increase the probability of translocation via OmpF, instead of direct LPS permeation. While the typical translocation of the three AMPs through the OmpF pore required a maximum force of ~2500 pN and a considerable displacement of the constriction loop (L3), it was possible to translocate the P6m peptide with a force of only ~800 pN if the interaction with the constriction loop (L3) was kept to a minimum. This finding suggested that the maximum force required to translocate the three peptide variants across the OmpF pore is not a function of amino acid sequence, in the linear configuration used for SMD simulations, but rather is dependent on the position of the L3 loop inside the OmpF lumen. Also indicated by the fact that translocation of protamine, a peptide of more than three times the molecular mass of P6 was translocated via CymA from *Klebsiella oxytoca* with a similar force requirement (maximum ~ 900 pN) [13]. The dimensions of the constriction region of OmpF are 7 × 11 Å, resulting in a solvent-accessible area of ~32 Å^2^ [42], but it was found that the intrusion of loop 3 into the permeation pathway during MD simulations can reduce the area to 15–19 Å^2^ [43,44]. Thus, the translocations of three AMPs that produce extensive displacement of loop L3 by ~3 Å and a force of well over 2000 pN, are probably less feasible in vitro.

## 4. Materials and Methods

### 4.1. MD Simulations

In order to investigate the hypothetical mechanisms which could be employed by AMPs to permeate a Gram-negative OM, two versions of a membrane model representative of *E. coli* were constructed with the lower leaflet consisting of the following phospholipids: 1-palmitoyl-2-palmitoleoyl-sn-glycero-3-phosphoethanolamine (PPPE), 1-palmitoyl-2-vacenoyl-sn-glycero-3-phosphoglycerol (PVPG) and 1,1′-palmitoyl-2,2′-vacenoyl cardiolipin (PVCL2) in 75:20:5 molar ratio [45]. The upper layer of the first membrane model was composed exclusively of smooth LPS (S-LPS) molecules with the following structure: type 1 lipid A, K12 core oligosaccharide and O21 serotype O-antigen with a single polysaccharide unit. All *E. coli* K12 strains are phenotypically rough, with a complete K12 core, but lacking O-antigen polysaccharide, although the *E. coli* strain K12 MG1655 can be used to restore or control O-antigen production [46,47,48]. The second *E. coli* membrane model, with identical lower leaflet configuration, and with the upper leaflet composed of rough LPS (R-LPS): type 1 lipid A and R1 core oligosaccharide, but lacking any O-antigen polysaccharide. Two additional molecular systems based on the S-LPS *E. coli* membrane model with a single O21 O-antigen serogroup unit, but without the monomeric OmpF channel, were constructed. One was used to test P6 peptide interaction with a continuous LPS layer, i.e., in the absence of the gap created by a transmembrane protein, and the other was used as a reference, i.e., to analyze LPS and phospholipids dynamics in the absence of OmpF channel and AMPs.

The molecular systems setup was automated with an Apache Taverna [49] workflow that integrated CHARMM-GUI [50] LPS Modeler [51] input scripts used to build the LPS layer, Membrane Builder [52,53] input scripts used to build the phospholipid (PL) layer, with custom CHARMM [54] scripts for building AMPs from an amino acid sequence, arrangement, and placement within the system. Two layers of 2 × 3 AMPs (12 total molecules of either P6, P6m or Pxm) were placed on top of the LPS-PL bilayers, facing the LPS side, at a height of 5 and 15 Å, respectively. A single monomer of *E. coli* OmpF porin (PDB ID 4LSE) [55] was embedded in LPS-PL bilayers and oriented by aligning the first principal of OmpF along the membrane z axis. The molecular systems were solvated with the TIP3P water model and a physiological ionic concentration (K^+^ and Cl^-^) of 150 mM/L was set. Lipid A and the core oligosaccharide of 36 LPS molecules were neutralized with Ca^2+^ ions. The final system size along the z axis was set by the addition of bulk water with a thickness of 40 Å to the upper leaflet and 20 Å on the lower leaflet. Additional details pertaining to all atomistic molecular systems are provided in Appendix A. The completed molecular systems were minimized, gradually heated to 300 K, and a pressure of 1 bar was set. The systems were relaxed following the CHARMM-GUI 6-step protocol, i.e., gradual reduction of planar and dihedral restraints applied to LPS, PL, protein (OmpF and AMPs), and water molecules. The production NPT simulations between 450–1000 ns were run with NAMD v2.12 [56] using CHARMM36m force field [57]. Langevin dynamics was used to maintain a constant temperature, with a coupling coefficient of 1 ps^- 1^. Constant pressure was maintained with Nosé-Hoover Langevin piston [58], using a piston period of 50 fs and a decay time of 25 fs. Periodic conditions were applied, and particle mesh Ewald was used for long-distance electrostatic interactions [59]. Van der Waals interactions were attenuated at a distance of 10–12 Å, while hydrogen bonds were fixed at an ideal distance using the SHAKE algorithm with 2 fs time steps [60].

Area per lipid analysis was performed using GridMAT-MD [61]. Graphical representations were prepared using VMD [62].

### 4.2. Potential of Mean Force (PMF)

Potential Mean Force (PMF) was used to estimate the free binding energy of antimicrobial peptides to the two membrane models. For the calculation of PMF, two sets of 10 SMD (steered molecular dynamics) non-equilibrium pulls were performed in opposed directions: forward (*fwd*), in which AMPs are returned in the initial binding position in the membrane model, and reverse (*rev*) in which AMPs are pulled from the initial binding position in the membrane model. The z axis was used as the reaction coordinate for all steered MD simulations. AMPs center of mass was used to pull the molecules for a distance of 45 Å. The *fwd* pulls were prepared by pulling the AMP, from the initial binding position, to the target distance, followed by 50 ns of NPT equilibration. The *rev* pulls were also prepared by 50 ns of NPT equilibration of AMPs in the initial binding configuration. Harmonic constraints were used to limit the lateral diffusion of LPS that come into contact with AMPs in the binding configuration. Ten snapshots were extracted from each of the two MD simulations, at 5 ns intervals, to produce the pulling configurations. The SMD pulls were performed with a harmonic constraint force constant or spring constant (*k*) of 7 kcal/mol/Å^2^, at a rate of 0.0045 Å/ps. PMF at equilibrium along the reaction coordinate was reconstructed using the average external mechanical work (*W*) made during the non-equilibrium of *fwd* and *rev* pulls [63].

Three to four of the initial peptide configurations bound to the LPS layer (*rev* pulls), employed for PMF calculations, were reused in order to evaluate the force requirement of the three AMPs to translocate the OmpF pore. Preliminary SMD pulls were required for P6m and Pxm peptides in order to align (on x and y coordinates) either the N- or C-terminal region with the OmpF extracellular opening. The spring constant was adjusted to the lowest value that allows the peptides to pass through the pore in a linear configuration (k = 0.15 kcal/mol/Å), in order to minimize the strain on the OmpF secondary structure.

## 5. Conclusions

In summary, the interaction of P6, P6m, and Pxm antimicrobial peptides with *E. coli* LPS structures, in particular with the phosphate groups of the inner core oligosaccharide, accessed through the opening in the LPS layer left by the OmpF, may be the first step in OM permeation. However, we believe that the binding of these three AMPs to the putative LPS binding site, such as P6m in the S-LPS *E. coli* membrane model, is not sufficient for OM translocation at the protein-lipid interface, as in the case of colicin N, or by direct permeation. Instead, the binding of the AMPs to the LPS inner core oligosaccharides, in the proximity of the OmpF pore opening, could increase the probability of translocation via OmpF in a linear or “un-folded” configuration. Translocation of any molecule across OmpF porin is highly dependent on the constriction loop (L3), which significantly reduces the diameter of the pore. As shown by the steered MD simulations results, the position of loop L3 inside the OmpF lumen can be less obstructive, leading to less engagement with the P6m peptide and requiring only a fraction of the typical pulling force for translocation. Unlike colN, the P6 peptide and the two derivatives do not have a molecular weight or a folded configuration that prevents their translocation by the OmpF porin, but in vitro tests are required to test this hypothesis.

## Figures and Tables

**Figure 1 ijms-24-02005-f001:**
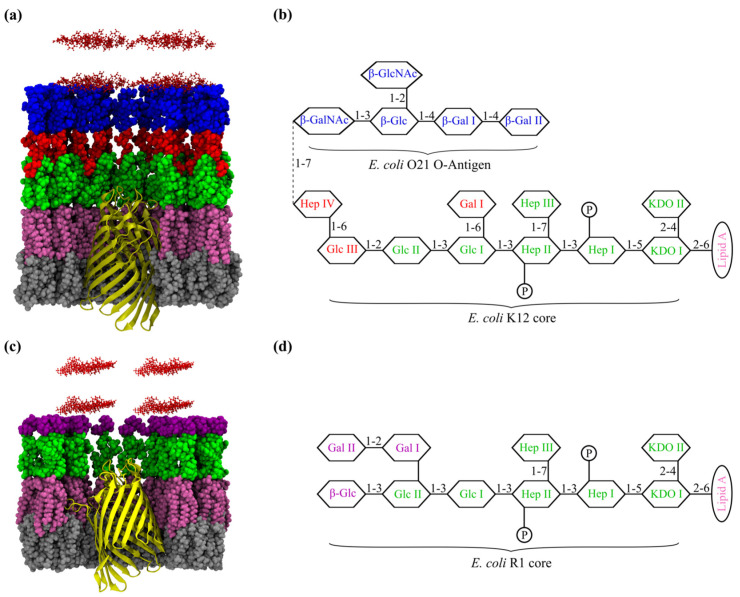
Representation of the major types of molecular systems analyzed in this paper: (**a**) cross-section of OmpF inserted in the S-LPS *E. coli* (Lipid A, K12 core, 1 O21 O-antigen) membrane model and (**c**) cross-section of OmpF inserted in the R-LPS *E. coli* (LipidA, R1 core) membrane model; (**b**) sequence of S-LPS and (**d**) sequence of R-LPS used to build the two distinct *E. coli* membrane models. Lipid A is colored in mauve; shared residues between the two LPS types are colored in green; the unique saccharide residues of the two core types are colored in red and violet; O-antigen is colored in blue; while PL from the bottom leaflet are colored in gray.

**Figure 2 ijms-24-02005-f002:**
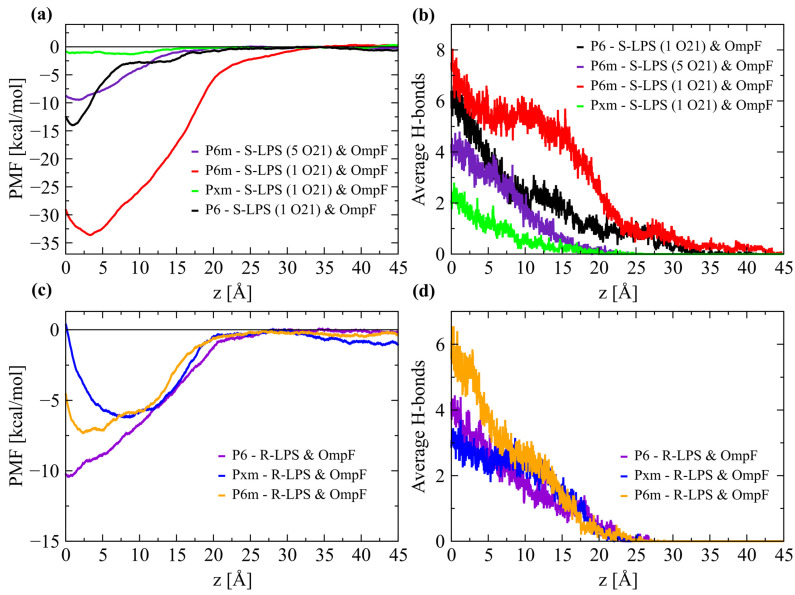
PMF binding energy of representative AMPs inserted in (**a**) S-LPS and (**c**) R-LPS *E. Coli* membrane models. Corresponding hydrogen bonds formed between the AMPs and LPS molecules/OmpF porin, averaged across all non-equilibrium MD simulations used for PMF calculation of (**b**) S-LPS and (**d**) R-LPS systems.

**Figure 3 ijms-24-02005-f003:**
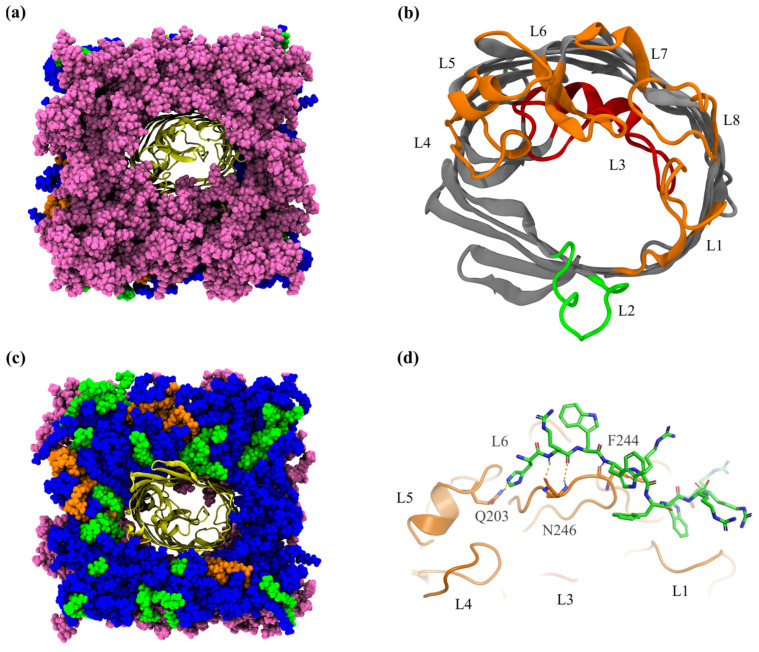
Graphical representation of OmpF inserted in the *E. coli* OM composed of R-LPS: (**a**) top view of extracellular side and (**c**) periplasmic side (OmpF is colored in yellow, LPS in mauve, PPPE in blue, PVPG in green, PVCL2 in orange). (**b**) Location of OmpF key loops and (**d**) the interaction of P6m peptide (stick representation) with L5 and L6 loops of OmpF.

**Figure 4 ijms-24-02005-f004:**
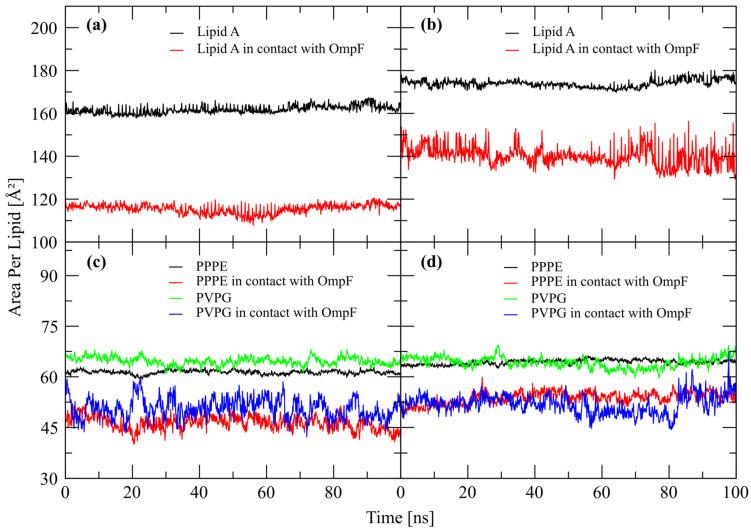
Lateral packing of LPS (lipid A) and PL (PPPE and PVPG) molecules by the OmpF monomer during MD simulations: (**a**) lipid A from S-LPS, (**b**) lipid A from R-LPS systems; (**c**) PL from S-LPS and (**d**) PL from R-LPS systems.

**Figure 5 ijms-24-02005-f005:**
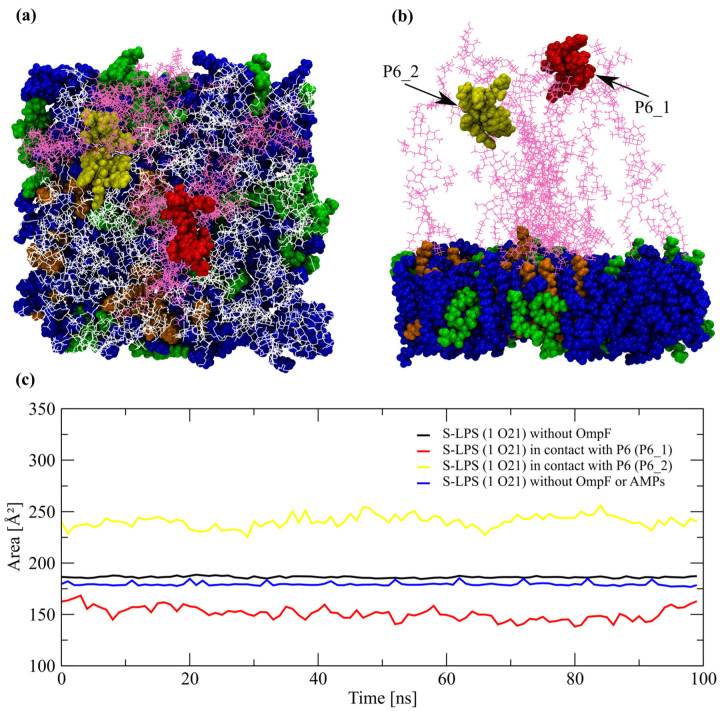
Graphical representation of two P6 peptide molecules inserted in a S-LPS *E. Coli* membrane model without the OmpF monomer: (**a**) top view and (**b**) side view. LPS chains in contact with the two P6 molecules are colored in mauve, while the rest are colored in white, PPPE are colored in blue, PVPG are colored in green, PVCL2 are colored in orange. (**c**) APL-like analysis of regions of the LPS layer interacting with P6 peptide, one with relatively higher density of O-antigen molecules (P6_1) and a second with relatively lower density of O-antigen molecules (P6_2).

**Figure 6 ijms-24-02005-f006:**
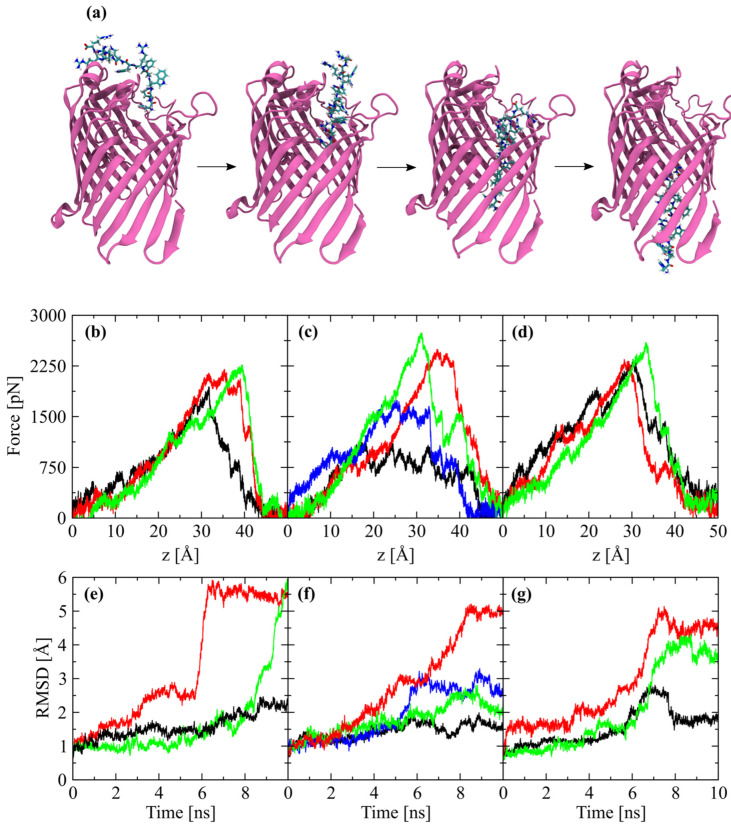
Translocation of AMPs through OmpF pore. (**a**) Representative snapshots of P6 peptide pulled through the OmpF pore using a constant-velocity steered MD simulation. Force profiles as a function of the reaction coordinate *z* for: (**b**) three repeated P6 SMD pulls, (**c**) four repeated P6m SMD pulls and (**d**) three repeated Pxm SMD pulls through OmpF pore. RMSD analysis of the constriction loop (L3) for the corresponding SMD pulls: (**e**) P6, (**f**) P6m and (**g**) Pxm peptide.

**Table 1 ijms-24-02005-t001:** Physical and chemical properties of AMPs.

AMP	Hydrophobicity (%)	Net Charge *	Molecular Weight (Da)
P6—HRWWRWWRR-NH_2_	44.44	+4.1	1524.76
P6m—HRWWRWWRRR-NH_2_	40	+5.1	1680.95
Pxm—HWRRWRRW-NH_2_	37.5	+4.1	1338.55

* At pH 7.0.

**Table 2 ijms-24-02005-t002:** Comparison of AMPs binding energy to *E. coli* OM with embedded OmpF porin.

System	PMF (kcal/mol)	z * (Å)
P6—R-LPS	−10.41	−0.57
P6m—R-LPS	−6.21	−0.11
Pxm—R-LPS	−7.33	−2.74
P6—S-LPS, 1 O21	−13.99	−10.66
P6m—S-LPS, 1 O21	−33.61	−12.32
P6m—S-LPS, 5 O21	−9.44	7.67
Pxm—S-LPS, 1 O21	−1.33	8.12

* Distance relative to the LPS surface at which the binding energy minima was achieved.

## Data Availability

All simulation data are available on request.

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
