# Peer review of "Interaction of Tryptophan- and Arginine-Rich Antimicrobial Peptide with E. coli Outer Membrane—A Molecular Simulation Approach"

_ijms, 2023, doi:10.3390/ijms24032005_

Round 1

Reviewer 1 Report

The manuscript is very well written and done as per scientific norms. The overall data presented in the paper is appreciated and the figures quality is statisfactory. 

Thought the simulation time is enough, some additional time might be significant. I will advice to increase the simulation time a bit. 

Limitations of the work must be presented in the revised version. 

Author Response

We thank the reviewers for the thoughtful critiques of our manuscript and for their suggestions. We considered all of the reviewers’ comments and corrected our paper accordingly.

O1) Thought the simulation time is enough, some additional time might be significant. I will advice to increase the simulation time a bit.

Response

Depending on the scope of the study is possible to obtain more information by extending the simulation time, however, we believe that the simulation time is sufficient for the aim of this study.  We assessed the simulation time requirement by comparing the atomistic AMPs insertion results with the Coarse-Grained (CG) ones (Supplemental information) i.e., an extended simulation of 145 micro seconds - 145 times longer than atomistic simulations, which required ~12 months to complete - and we concluded that the simulation time for atomistic systems is appropriate given the ratio of insertion events per simulation time unit between the two methods.

O2) Limitations of the work must be presented in the revised version.

Response

Although every study has limitations, we do not consider the simulation time a limitation, per se. The methods employed in this study are state of the art and are used in numerous publications [1-3]. Also, the cited studies used NPT production simulations between 200 - 450-ns.  However, following the reviewer suggestion, we addressed the limitation of not exploring the insertion of AMPs in the Lipid A - PL medium (discussion pertaining to the scrambled Pxm peptide, at page 13).

Reviewer 2 Report

The manuscript entitled “Interaction of an tryptophan- and arginine-rich antimicrobial peptide with E. coli outer membrane - a molecular simulation approach” by Necula et al talk about the interaction of

tryptophan and arginine-rich peptides with two different LPS types namely (i)smooth LPS composed of lipid A, K12 core and O21 O-antigen, and (ii)rough type LPS composed of lipid A and R1 core.

The work is interesting and gives a very detailed insight

into the interaction aspect of the AMP and LPS. However, just when the interest in the manuscript develops, it finishes up.

I would like to hear some discussion and the

"significance" aspect of the work from the authors. Hence I request

them to revise on the lines of the following comments.

Specific comments below:

1) Please make a Table of all the peptides used in the

study and mention the Net Charge, Percent Hydrophobicity and molecular weight of the peptides.

2) Tryptophan and the arginine-rich peptides have long

been known to bind to the LPS and inhibit LPS-mediated toxicity. Can the

authors write something about the exclusive role of arginines in LPS binding?

Can the same be achieved by using Lysines in place of Arginines? If the results

are the same for both arginines and lysines, then I would request the authors to do the MD simulation with the scrambled peptides to address the importance of arginines

3) Please mention the medical aspect of LPS binding to

AMPs. As the authors might be aware that sepsis has no drug, can they discuss something about how their work could contribute to understanding sepsis and targeting it? For theory about AMP and LPS interaction, you may read or refer the following article”

 Selective phenylalanine to proline substitution for

improved antimicrobial and anticancer activities of peptides designed on

phenylalanine heptad repeat. 

Acta Biomaterialia

Volume 57, 15 July 2017, Pages 170-186

4) Remove the article "an" from the title.

Interaction of an tryptophan- and arginine......approach

Author Response

We thank the reviewers for the thoughtful critiques of our manuscript and for their suggestions. We considered all of the reviewers’ comments and corrected our paper accordingly.

O1) Please make a Table of all the peptides used in the study and mention the Net Charge, Percent Hydrophobicity and molecular weight of the peptides.

Response

A table with AMPs properties was added at page 3 of the revised manuscript.

O2) Tryptophan and the arginine-rich peptides have long been known to bind to the LPS and inhibit LPS-mediated toxicity. Can the authors write something about the exclusive role of arginines in LPS binding? Can the same be achieved by using Lysines in place of Arginines? If the results are the same for both arginines and lysines, then I would request the authors to do the MD simulation with the scrambled peptides to address the importance of arginines

Response

We believe that the importance of Arg residue is highlighted by the overwhelming role in LPS binding, even for Pxm derivative, although it is not exclusive (addressed at page 13). Although Lys and Arg may have the same charge and similar structure, we do not expect the same results with Lys, mainly because of the number of hydrogen bond donors: Lys has 3 while Arg has 5. In some instances, Lys cation-π interactions can be stronger than Arg, but it is diminished by the surrounding dielectric medium, becoming essentially negligible in strength and without well-defined equilibrium separation [4]. In the stacked conformation, the Arg sidechain is able to form almost as many hydrogen bonds with the surrounding water molecules as when it is not involved in any cation–π interactions [5]. This is in contrast to lysines, which cannot form hydrogen bonds while engaged in cation–π interactions with an aromatic residue [5, 6]. It was suggested that intramolecular cation–π interaction of Arg with Trp residues essentially shields Arg from the highly hydrophobic nature of the bilayer [7].

O3) Please mention the medical aspect of LPS binding to AMPs. As the authors might be aware that sepsis has no drug, can they discuss something about how their work could contribute to understanding sepsis and targeting it? For theory about AMP and LPS interaction, you may read or refer the following article ”Selective phenylalanine to proline substitution for improved antimicrobial and anticancer activities of peptides designed on phenylalanine heptad repeat. ” Acta Biomaterialia Volume 57, 15 July 2017, Pages 170-186

Response

Thank you for the suggestion. This point was addressed in discussion at page 13. Unfortunately, tryptophan- and arginine-rich AMPs do not seem to have effective neutralization activity, but my serve as future leads for improved AMP designs.

O4) Remove the article "an" from the title

Response

We removed the "an" article from the title.

Round 2

Reviewer 2 Report

The authors have done a very good job of revising the manuscript. I recommend publishing this work in its present form.

It was a pleasure reviewing their work.